# Characterizing the quantum field theory vacuum using temporal Matrix Product States

Emanuele Tirrito[1], Luca Tagliacozzo[2,3*], Maciej Lewenstein[1,4] Shi-Ju Ran[5,1*]

**1** ICFO-Institut de Ciencies Fotoniques, The Barcelona Institute of Science and Technology, 08860 Castelldefels (Barcelona), Spain
**2** Department of Physics and SUPA, University of Strathclyde, Glasgow G4 0NG, United Kingdom
**3** Departament de Física Quàntica i Astrofísica and Institut de Ciències del Cosmos (ICCUB), Universitat de Barcelona, Martí i Franquès 1, 08028 Barcelona, Catalonia, Spain
**4** ICREA, Passeig Lluis Companys 23, 08010 Barcelona, Spain
**5** Department of Physics, Capital Normal University, Beijing 100048, China
* luca.tagliacozzo@strath.ac.uk
* ranshiju10@mails.ucas.ac.cn

October 30, 2018

## Abstract

**In this paper we construct the continuous Matrix Product State (MPS) representation of the vacuum of the field theory corresponding to the continuous limit of an Ising model. We do this by exploiting the observation made by Hastings and Mahajan in Phys. Rev. A 91, 032306 (2015) that the Euclidean time evolution generates a continuous MPS along the time direction. We exploit this fact, together with the emerging Lorentz invariance at the critical point in order to identify the matrix product representation of the quantum field theory (QFT) vacuum with the continuous MPS in the time direction (tMPS). We explicitly construct the tMPS and check these statements by comparing the physical properties of the tMPS with those of the standard ground MPS. We furthermore identify the QFT that the tMPS encodes with the field theory emerging from taking the continuous limit of a weakly perturbed Ising model by a parallel field first analyzed by Zamolodchikov.**

# 1   Introduction

In this paper we focus on the continuous limit of a lattice model using tensor networks. Tensor networks allow to encode equilibrium states of many-body quantum systems described by local Hamiltonians with modest computational resources [1–15]. They can be used to encode both quantum states, or to represent classical partition functions as a result of the well-known classical to quantum correspondence, based on the path-integral formulation of quantum mechanics.

The success of tensor networks started with White's DMRG algorithm that, as of today, is still the best numerical tool to characterize strongly-correlated systems in 1D [16–20]. They soon were extended to higher dimensions and connected with the theory of entanglement in many-body quantum systems that was being developed in parallel by the quantum information community [21–33].

In recent developments, tensor network have been used to study continuous quantum field theories (QFT). Rather than starting from a lattice model, one can indeed start from a continuous Hamiltoninan describing a QFT and formulate a variational calculation in order to express the vacuum of the QFT as tensor network [34–44].

Here we extract the matrix product state of a (1+1)D QFT vacuum by starting from a lattice model and constructing explicitly its continuous limit. The standard way for doing this is to reduce the lattice spacing while approaching a quantum critical point as customary in the context of building the continuous limit of lattice field theory [45]. This entails two limits, the lattice spacing $a$ going to zero, and the correlation length $\xi$ going to infinity as we approach the critical point. The two limits needs to be taken in parallel, by keeping their product $a\xi$ constant fixed to the desired physical value.

In the context of tensor networks, this procedure can be considerably simplified by observing that, in order to extract the ground state of a lattice system, we usually perform an imaginary time evolution. A $D$ dimensional system thus actually becomes $D + 1$ dimensional once we include the imaginary time. The evolution along the imaginary time is performed by first breaking it in small time steps. Each small step is then approximated by using a Suzuki-Trotter decomposition [46, 47]. The decomposition becomes more and more accurate as the small time steps tend to zero. Reducing the time steps, thus can be seen as analogous to taking the continuous limit along the time direction, something that Hastings had already observed in Ref. [48].

This observation can be exploited in order to extract a continuous MPS along the time direction that we call tMPS. The standard QFT vacuum to vacuum transition probabilities can thus be expressed as the norm of the tMPS state. Beside this fact, in general the tMPS encodes a different state from the ground state MPS of the discretized field theory that can be represented as a standard MPS by, e.g., performing the imaginary time evolution of a random initial MPS for very long times.

If the system possesses a Lorentz-invariant critical point, however, the tMPS and the standard ground state MPS, at the critical point, are related by a trivial non-universal re-scaling factor. In these specific cases, the tMPS thus encodes the continuum limit of the ground state of the discretized field theory, that is the tMPS represents the vacuum of the QFT.

Here we will check this statement. In particular we want to better characterize the regularized QFT described by the tMPS. The main advantage of encoding the vacuum of a QFT with a tensor network with finite bond dimension is indeed that the QFT is then automatically regularized. For example, the entanglement entropy of half of the vacuum is finite. The bond dimension of the tensor network thus acts as the UV regulator of the field theory, by limiting the number of degrees of freedom per unit length. This was first observed in [22, 34].

Despite the numerous studies that followed this initial observations [50–57] we still lack a full understanding of the role of the bond dimension as a regulator.

In particular, it is well known that, starting from a lattice model, the low-energy spectrum of the emerging field theory depends on both the critical exponents at the quantum critical point and the boundary conditions of the lattice system [58, 59]. This fact translates into well defined ratios for the masses of the low energy excitations only depending on the universality class of the quantum-phase transition. For example, by taking

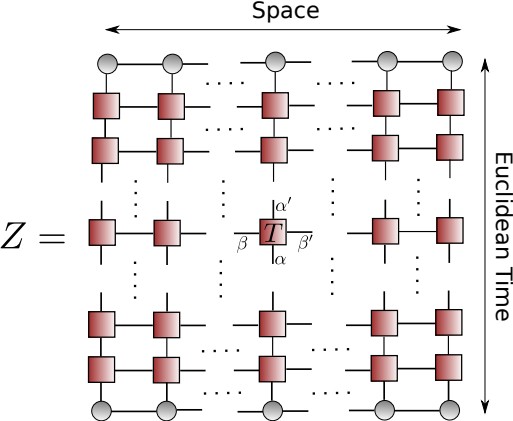

Figure 1: (Color online) Representation of 2D tensor network encoding the vacuum to vacuum transition probability corresponding to the norm of the ground state obtained by performing an imaginary time evolution of an initial matrix product state. Geometric shapes represent the elementary tensors, and the lines encode their contraction.

the continuous limit of the Ising model on a 2D torus, we obtain the Ising field theory whose low-energy excitations have masses fulfilling $m_2/m_1 = 4$ [49, 61, 62].

Since the tMPS transfer matrix propagates the dynamical correlations, one would naively expect to recover these ratios from appropriate functions of the eigenvalues of the tMPS transfer matrix. Our analysis of the tMPS for the Ising model show that this does not happen. Interestingly we find that the masses we extract from the tMPS correspond to those predicted by Zamolodchikov, $m_2/m_1 = 1.6$. These masses were obtained when characterizing the field theory emerging from the critical Ising model perturbed with a small parallel field [60, 62].

While this result could seem surprising, it can be easily understood by noticing that the finite bond dimension of a tMPS breaks the $Z_2$ symmetry in a similar way than the magnetic field does in the Zamolodchikov's scenario. The finite bond dimension thus acts as a relevant magnetic perturbation to the critical Hamiltonian, rather than as a thermal perturbation, and the spectrum is thus modified accordingly.

These results agree with what was discussed in [53]. There the authors compared the finite-size scaling implicit by taking a continuous limit of a discrete lattice model, and calculated the finite-entanglement scaling that we also use here to obtain the tMPS. They found that the two procedures generate different fixed point tensors. In performing the finite-size scaling with an MPS, indeed, we first fix the size of the system to a finite value $N$ and increase the bond dimension $\chi$ until we accurately encode the ground state of the finite-size system. This procedure corresponds, in the MPS language, to taking first $\chi \to \infty$ and then $N \to \infty$.

The tMPS is obtained by inverting the order of the two limits, we first take $N \to \infty$ with fixed $\chi$, and then $\chi \to \infty$. As observed in [53] the two limits do not commute. At any finite size (either spatial or temporal) indeed the $Z_2$ symmetry is restored, while at any finite $\chi$ the $Z_2$ symmetry is broken.

In this paper we focus on the critical point of Ising model in a transverse field and derive these results by first showing that the tMPS is actually a continuous MPS [26]; it describes the same physics as the ground-state MPS up to a gauge transformation and some trivial rescaling factors. This allows us to analyze the low-energy spectrum of the field theory emerging from the tMPS and identify it with the one predicted by Zamolodchikov et al [60, 62].

## 2   Basic definitions and preliminaries

We want to evaluate a 2D TN with the structure shown in Fig. 1. The TN will be contracted to a scalar $Z$ whose value could represent different physical quantities, depending on the content of the individual tensors.

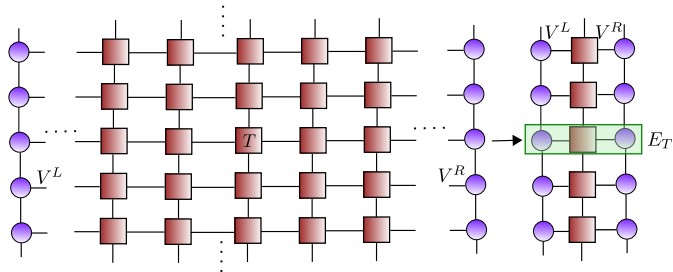

Figure 2: (Color online) Temporal Matrix Product State (tMPS). The MPS fixed point obtained by contracting the 2D TN of Figure 1 from left to right defines the tMPS. It describes a state along the Euclidean time direction and becomes a continuous MPS in the limit of the Trotter step going to zero.

For example $Z$ could encode the partition function of a $2D$ classical model, the norm of a 2D quantum state encoded in a PEPS, or the real or imaginary time evolution of a 1D quantum state. In our paper $Z$ will encode the latter case, the imaginary time evolution of a 1D quantum state $|\psi\rangle$.

The $1D$ system is made by $N$ constituents whose interaction is described by, for simplicity, a nearest neighbors Hamiltonian $\hat{H} = \sum_n \hat{H}^{[n,n+1]}$. $\hat{H}^{[n,n+1]}$ encodes the two-body interactions. If a system is translationally invariant, as in the cases we will consider here all the two body terms are the same, namely $\hat{H}^{[n,n+1]} = \hat{H}^{[2]}$.

The imaginary time evolution, performed for sufficiently large times, allows to approximate the ground state of $\hat{H}$ [1],

$$|\Omega\rangle = \lim_{\beta \to \infty} \frac{e^{-\frac{\beta}{2}\hat{H}}|\psi\rangle}{||e^{-\frac{\beta}{2}\hat{H}}|\psi\rangle||}. \tag{1}$$

Alternatively the ground state of $\hat{H}$ could be obtained variationally by minimizing the energy over the class of normalized MPS states with fixed bond dimension,

$$|\Omega\rangle = \arg\min_{|\psi\rangle}\left\{\langle\psi|\hat{H}|\psi\rangle\right\}. \tag{2}$$

$|\psi\rangle$ is an MPS state fulfilling $\langle\psi|\psi\rangle = 1$.

At large fixed $\beta$ in eq. (1) we need to divide the evolution in small steps by fixing $M$ such that $\beta/2M = \tau \ll 1$. In this way we can approximate Eq. (1) step by step using a Suzuki-Trotter decomposition at the chosen order in $\tau$.

When $|\psi\rangle$ is in an MPS form, each step $U(\tau)|\psi\rangle \equiv e^{-\tau\hat{H}}|\psi\rangle$ can be performed approximately by first contracting the TN and then truncating the MPS back to desired bond dimension D after normalizing the state (see the Appendix A.1 for more details). The fixed point of this procedure provides the MPS representation of $|\Omega\rangle$.

As a result, in the case of the imaginary time evolution, the individual tensors of the 2D TN represented in Fig. 1, are related to the small steps of time evolution $U(\tau)$ (see appendix A.3).

The length of horizontal direction encodes the number of constituents $N$ of the $1D$ quantum system, and we call it in the following spatial direction. The vertical length encodes the number of Trotter steps $M$ and we call it Euclidean-temporal direction.

Looking at the $2D$ TN we can envisage a different contraction scheme (see Ref. [48, 63]). Rather than contracting the TN downwards, we can contract it from left to right as in (Fig. 2). Once again, if we enforce an MPS structure for the contraction (alternating contraction and truncation steps) we can define the tMPS as the fixed point of this contraction strategy,

$$|tMPS\rangle = \sum_{\{\beta\}} \cdots V_{\alpha_i\beta_i\beta_{i+1}} V_{\alpha_j\beta_j\beta_{j+1}} \cdots |\cdots \alpha_i \alpha_j \cdots\rangle. \tag{3}$$

---
[1] We assume that $\langle\Omega|\psi\rangle \neq 0$ as expected for a randomly chosen $|\psi\rangle$.

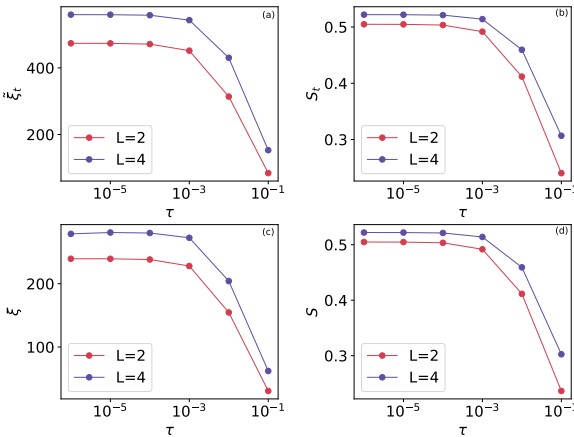

Figure 3: *Correlation time* **(a)**. The physical correlation time $\tilde{\xi}_t$ versus $\tau$. We appreciate that $\tilde{\xi}_t$ stays finite when $\tau$ goes to zero. The system is at the critical point $h = J$ where we expect the correlation time to diverge. However, $\tilde{\xi}_t$ stays finite as a result of the finite bond dimension of the tMPS. *Entanglement entropy of the tMPS* **(b)**. The entanglement entropy of the tMPS only depends on the physical correlation time $\tilde{\xi}_t$ and stays finite when $\tilde{\xi}_t$ is finite. At the critical point the linear dispersion relation of the low-energy excitations implies an enhanced space-time symmetry. This can be checked by studying the correlation length in the ground state $\xi$. Once more, it is finite as a result of the finite bond dimension $\chi$. The entanglement entropy $S$ of half of the ground state also stays finite, and weakly depends on $\tau$ as expected. We take the length of the unit cell $L = 2, 4$ and the bond dimension cut-off $\chi = 20$.

The dots encode the fact that the tMPS is infinite ($\beta$ and $M$ diverge), and $V$ denote its constituent tensors. Notice that the quantum sate is now defined as a state with fixed "auxiliary" position and one constituent per Trotter step, thus effectively encoding the time evolution of a single coarse-grained constituent. In the next sections, we characterize the tMPS through the AOP algorithm [64, 65] (described in the Appendix A.4), where the tensor $V$ is obtained by solving a set of self-consistent eigenvalue problems.

## 3   The continuous tMPS

Besides very specific scenarios, the fixed points extracted after the contraction along the spatial and the temporal directions are different. The lattice spacing along the horizontal direction is discrete, while, in the limit $M \to \infty$ it becomes continuous in the vertical direction. While the parallel MPS represents the ground state of $\hat{H}$ the tMPS sites encode the different instant of time of the evolution of a single coarse-grained constituent. Since in the limit $M \to \infty$ the time step goes to zero, the constituent continuously varies in time, and thus the tMPS encodes a continuous system [34].

When the $2D$ TN represents a classical isotropic model, the fixed points obtained by contracting the network along either the vertical or the horizontal directions should represent the same state. This equivalence has been tested on the TN that encodes the partition function of the $2D$ classical Ising model (see the Appendix C). This scenario can also occur in the quantum case in very special cases, where the original $\hat{H}$ possesses extra symmetries such as, e.g, the emerging Lorentz invariance we will consider in the following.

While the properties of the spatial MPS have been analyzed in several works, here we want to characterize the properties of the tMPS. This is a relatively unexplored area, and we only are aware of the results presented by Hasting and Mahajan [48].

We characterize the continuous nature of the tMPS by addressing the paradigmatic quantum Ising model

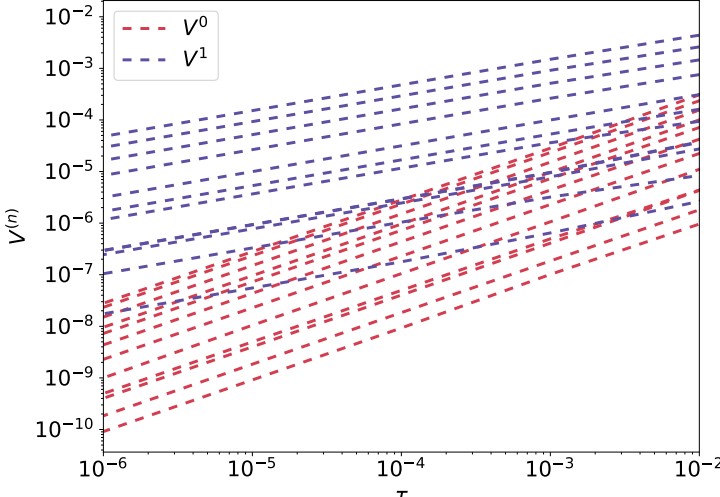

Figure 4: By plotting the elements of the tMPS tenors as a function of $\tau$ we can identify the elements whose extrapolation would produce $V^0$ and $V^1$ contribution in [Eqs. (7) and (8)]. They are distinguishable by the different slope of their $\tau$ dependence. We could thus reconstruct the $V^0$ and $V^1$ from a simple extrapolation of the finite $\tau$ data. These plots are a further confirmation that the tMPS converges to a continuous MPS.

defined by the Hamiltonian

$$H = J \sum_i S_i^x S_{i+1}^x - h \sum_i S_i^z, \tag{4}$$

with $S^x$ and $S^z$ are the Pauli matrices along the $x$ and $z$ directions. The model can be exactly solved [66,67] . The system is in a paramagnetic phase for values of the couplings such that $h/J > 1.0$. In the thermodynamic limit the order parameter acquires a non vanishing expectation value $\langle S^z \rangle \neq 0$ in the paramagnetic phase. In the ferromagnetic phase, for $h/J < 1.0$ the order parameter goes to zero. The two phases are separated by a quantum phase transition at $h/J = 1.0$, where the low-energy physics of the system can be described by a conformal field theory with central charge $c = 0.5$ [68–70].

The continuous nature of the tMPS emerges when taking the limit $M \to \infty$. If the system in the continuum has a finite correlation time $\tilde{\xi}_T$, we expect that the correlation time measured in terms of the lattice spacing $\xi_T$ should diverge in the limit as $\xi_T = \tilde{\xi}_T / \tau$. This is the first check that we perform on our tMPS since the correlation time of a translationally invariant tMPS is encoded in the gap of its transfer matrix $E_T \equiv \sum_\alpha V_\alpha \bar{V}_\alpha$,

$$\xi_T = \frac{1}{\ln \eta_0 - \ln \eta_1}, \tag{5}$$

where $\eta_n$ represents the $n$-th eigenvalue of $E_T$ (more details can be found in the Appendix B and Fig. 17).

This property is analyzed in Fig. 3. The correlation time $\xi_T$ obtained directly from the tMPS [Eq. (5)] diverges as expected as $\tau$ decreases. In panel (a) we show that, as expected, the physical correlation time $\tilde{\xi}_T$ converges rapidly when reducing $\tau$. In panel (b) we check that a physical quantity, the entanglement entropy of half of the temporal chain $S$, only depends on $\tilde{\xi}_T$. $S$ should indeed scale as $S \propto \log \tilde{\xi}_T$ [71].

Despite working at the critical point $h = J$ where the correlation time $\tilde{\xi}_T$ should diverge since $\hat{H}$ is gapless, we observe a finite correlation time. This is a consequence of describing the system with a finite bond dimension $\chi$, thus effectively cutting-off the correlations.

At the critical point, thanks to the linear dispersion relation of the low-energy excitations [67], we expect to observe an enhanced symmetry between space and time. Indeed, momentum plays the same role than energy, and we the theory becomes Lorentz invariant.

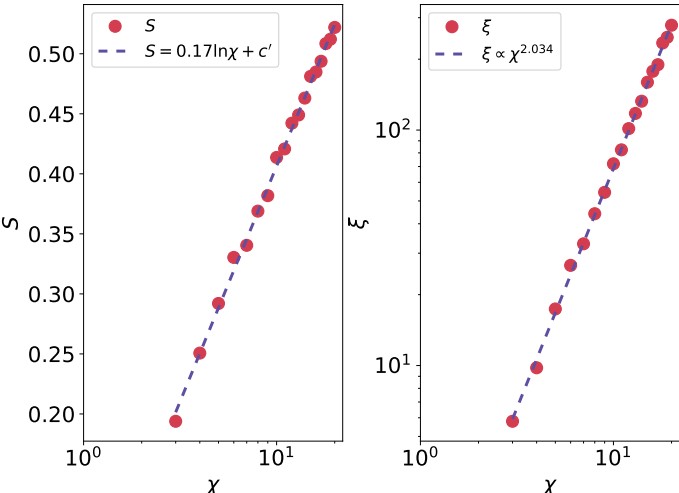

Figure 5: (Color online) —As a sanity check we reproduce the well known scaling of the correlation length $\xi$ and entanglement entropy $S$ versus the bond dimension cut-off $\chi$ of the spatial MPS. We take $h = 1$, and $\tau = 10^{-6}$.

In panel (c) and (d) of Fig. 3 we thus characterize the spatial correlation length $\xi$ and the entanglement entropy $S$ of half of the ground state $|\Omega\rangle$. In this setting $\tau$ only controls the accuracy of the Trotter expansion, and we thus expect the results to converge to a finite value as $\tau$ decreases. Once more both $S$ and $\xi$ stay finite at the critical point as a result of the finite bond dimension $\chi$. Our expectation are confirmed by the numerical results presented in panel (c) and (d) of Fig. 3. The dependence of the results on unit cell $L$ of AOP algorithm has been characterized in [64].

In order to further verify that the tMPS is continuous we compare its structure with the one of a cMPS. The continuous limit of an MPS was discussed in the context of the Bethe ansatz [72, 73]. In the tensor network community, the cMPS was first proposed by Verstraete *et al* [34]. The cMPS can be used as the variational state for finding ground states of quantum field theories, as well as to describe real-time dynamical features. The cMPS describes the low-energy states of quantum field theories once appropriately regularized, in the same way as a normal MPS describes the low-energy states of quantum spin systems. The cMPS can be constructed as the continuous limit of a discrete MPS defined as (see [34]),

$$|\psi\rangle = \sum_{n_1 \cdots n_L} V^{n_1} \cdots V^{n_M} \left(\Psi_1^\dagger\right)^{\sigma_1} \cdots \left(\Psi_M^\dagger\right)^{n_M} |\Omega\rangle, \tag{6}$$

where the $V$s satisfy

$$V^0 = I - \tau Q \tag{7}$$

$$V^1 = \tau R \tag{8}$$

$$V^n = \tau^n R^n \tag{9}$$

$$\Psi = \frac{\hat{a}_i}{\sqrt{\tau}}. \tag{10}$$

Both $R$ and $Q$ are independent from $\tau$.

We can thus use in particular (7) and (8) to extract $Q$ and $R$ for the Ising field theory, defined by the tMPS. In Fig. 4 we show the components $V^0$ and $V^1$ as a function of $\tau$. In the log-log plot we see that once we appropriately subtract the identity component to $V_0$, $V_0$ and $V_1$ scale differently as predicted by Eq. (7) and (8). This implies that by performing an appropriate scaling analysis, we can directly extract the $R$ and $Q$ for the tMPS of the Ising field theory. We only need to extrapolate the results at finite $\tau$ to the interesting $\tau \to 0$ limit, thus overcoming the difficulties that arise when trying to directly optimize the continuous MPS [34, 41, 52].

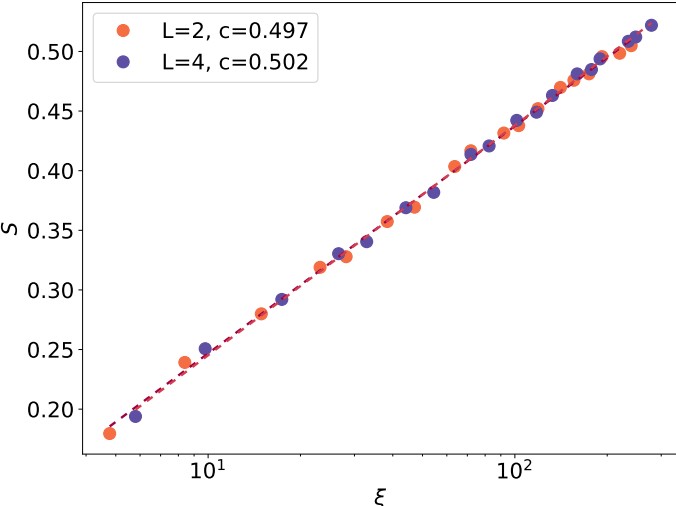

Figure 6: (Color online) The half chain entanglement entropy $S$ as a function of the correlation length $\xi$ at the critical point. Each point is obtained by chosing the bond dimension $\chi$ and the size of the unit-cell $L$. A fit to the expected logarithmic scaling allows to extract $c = 0.5$ as expected.

## 4 Identifying the emerging quantum field theory

A finite $\chi$ MPS at criticality induces a finite correlation length [22, 23, 52, 53] scaling as

$$\xi \quad \propto \quad \chi^{\kappa}. \tag{11}$$

This implies that the entanglement entropy $S$ scales as

$$S \quad = \quad \frac{c\kappa}{6} \ln \chi + c', \tag{12}$$

$$S \quad = \quad \frac{c}{6} \ln \xi + c'. \tag{13}$$

In order to characterize it, we use the AOP algorithm described in the Appendix A.4 and A.5. Beside the usual parameters, the algorithm can accommodate unit cells of different length $L$ [2]. Our results for the ground state MPS in Fig. 5 reproduce correctly Eq. (12).

In Fig. 6 we present the scaling of $S$ against $\xi$ of the spatial MPS with $L = 2$ and $4$ that allow to extract the central charge, whose value turns out $c = 0.502(5)$ as expected where the error is obtained by fitting several subsets of data for different $\chi$ and $L$.

Close to the critical point, the system becomes not only scale invariant but also Lorentz invariant (this is equivalent to the Galileo invariance in Euclidean time), as a consequence of the linear dispersion relation for low-energy excitations. For this reason it is possible to rotate the system and invert the role of space and time. In this way we can think of the tMPS as a state along the infinite temporal direction, and we expect it to share some properties (e.g. the scaling exponents) with the matrix-product state defined along the spatial direction.

We now characterize the correlation time $\tilde{\xi}_T$ and the entanglement entropy $S_T$ of the tMPS (that has been called temporal entanglement in Ref. [48]). As shown in Fig. 7 $\tilde{\xi}_T$ and $S_T$ also satisfy Eqs. (11) and (12), respectively. From our best fit, we extract $\kappa = 2.026(4)$ and $c_T = 0.504(3)$ for the tMPS compatible with the ground state data.

The anisotropic continuous limit introduced in the Trotter expansion, that only involves the temporal direction, induces a non trivial dependence between the physical quantities extracted from the MPS and the those

---

[2]The dependence of the results on $L$ has already been characterized in [64]. The dependence at criticality is also logarithmic. Here, unless stated otherwise, we use $L = 2$.

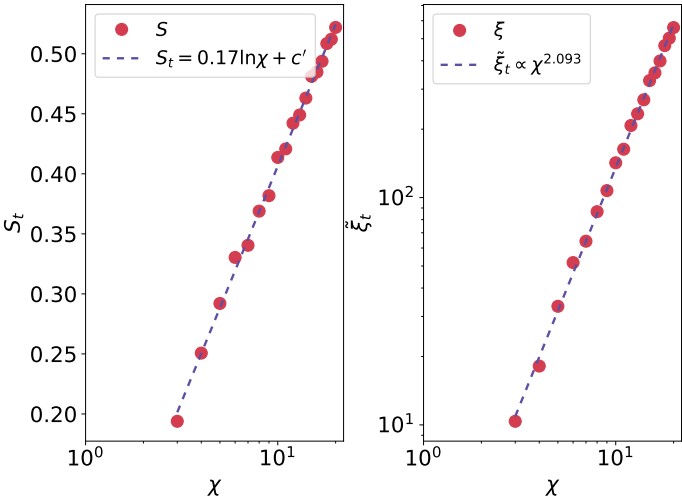

Figure 7: (Color online) The scaling of the correlation time $\tilde{\xi}_T$ and the temporal entanglement entropy $S_T$ versus the bond dimension cut-off $\chi$ of the tMPS. We take $h = 0.5$, $L = 2$ and $\tau = 10^{-6}$.

extracted from the tMPS. This can be understood to be the analogue of the well known physics of classical anisotropic models. They behave in the same way of their isotropic counterparts once non-universal rescaling factors are taken into account. For example the $2D$ anisotropic Ising model, where the coupling $J_s$ and $J_t$ along the two directions are different, possesses a line of critical points, all in the Ising universality class. They can be found from the equation $\sinh(2\beta J_t)\sinh(2\beta J_s) = 1$, where $\beta$ is the inverse temperature (see, e.g. [74]).

The rescaling factor $\nu$ can be extracted from the ratio of the correlation lengths along the two directions

$$\xi = \nu^{-1}\tilde{\xi}_T. \tag{14}$$

We can extract $\nu$ from our data in Fig. 8 and find it to be roughly $\nu = 2$.

We verify that the same factor lives between all the gaps of the temporal transfer matrix with the corresponding ones of the space transfer matrix.

We define

$$\tilde{\xi}_T^{(n)} = \frac{1}{\tilde{\Delta}_n} = \left(\log\frac{\tilde{\eta}_0}{\tilde{\eta}_n}\right)^{-1}, \tag{15}$$

$$\xi^{(n)} = \frac{1}{\Delta_n} = \left(\log\frac{\eta_0}{\eta_n}\right)^{-1}, \tag{16}$$

where $\eta$ and $\tilde{\eta}$ are the low-lying eigenvalues of the spatial and temporal transfer matrix. Note for $n = 1$, we have that $\tilde{\xi}_T^{(1)} = \tilde{\xi}_T$ and $\xi^{(1)} = \xi$.

In the same Fig. 8 we see that for all the $\chi$ we have considered, the gaps of the temporal transfer matrix are indeed proportional to the gaps of the spatial transfer matrix with the same proportionality constant $\nu = 2$. This fact confirms our interpretation of $\nu$ as a non-universal rescaling of the velocity of the excitations.

The tMPS is continuous but its entropy is finite. The bond dimension of the tMPS indeed plays the role of an UV regulator, as already observed in [34]. As expected, thus the state emerging in the continuum limit is a massive state, where the masses should be dictated by the Ising fixed point.

We thus would like to check this expectation by trying to identify the field theory corresponding to the continuous limit of the anisotropic model. One way to do this, is by characterizing the masses of the low energy excitations $m_1, m_2, m_3, m_4$. They can be extracted by constructing the appropriate ratios of the tMPS transfer matrix eigenvalues, namely $m_n = 1/\tilde{\xi}_T^{(n)}$. The masses we extract are in good agreement with those reported elsewhere [52].

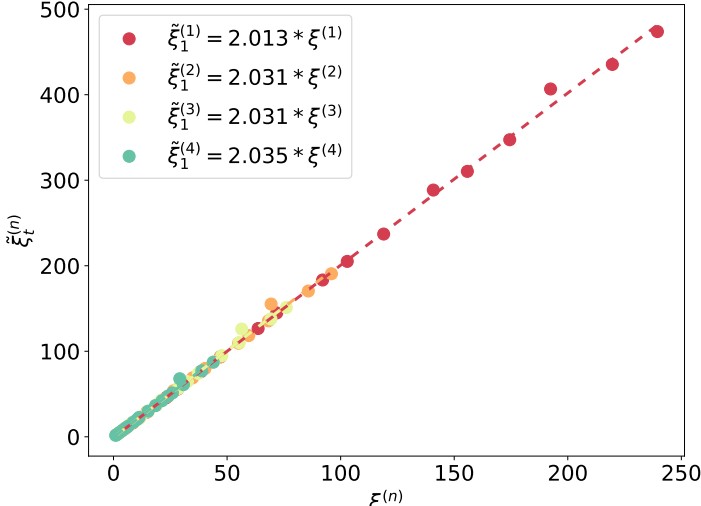

Figure 8: (Color online) Correlation length of the tMPS $\tilde{\xi}_T^{(n)}$ versus that of the spatial MPS $\xi^{(n)}$ for different bond dimensions $\chi$ and $\tau = 10^{-6}$. Our results show that $\tilde{\xi}_T^{(n)} = \nu\xi^{(n)}$ with $\nu = 2$.

The first mass is generally fixed by the rescaling of the Hamiltonian that ensures that the tMPS is normalized. This is an arbitrary choice. A more physical choice is to fix it to the value of the lowest mass of the field theory we expect to reproduce. We thus adjust $m_1$ in such a way that $m_2/m_1 = 1.618$. This is the ratio first proposed by Zamolodchikov [60] in his analysis of the Ising model weakly perturbed by a parallel field. The results of our numerical analysis of the mass ratios are presented in Fig. 9. There we can see that once we fix $m_2/m_1 = 1.618$, $m_3/m_1$ turns out to be and $m_3/m_1 = 2.00(5)$. This value is very close to the value of the second mass ratio predicted by Zamolodchikov $m_3/m_1 = 1.989$ [60].

Our masses thus seems to suggest that the continuous limit of an Ising MPS with fixed bond dimension $\chi$ reproduces the field theory obtained by perturbing the Ising model close to the critical point with a small parallel field of [60]. The above result can be understood by noticing that the finite bond dimension $\chi$ opens a gap similarly to what happens with a finite system size or a finite temperature. However, differently from the finite size or the finite temperature, the finite bond dimension also breaks the $Z_2$ symmetry.

Surprisingly our $m_4/m_1 = 3.65(23)$ results is compatible with the value of the ratio $m_7/m_1 = 3.891$ in the predictions by Zamolodchikov and not with the lightest mass ratios as we would have expected. Our identification thus still leaves a puzzle about why $m_4, m_5$ and $m_6$ are absent in the tMPS spectrum. We leave the solution to this puzzle to further analysis. We indeed believe it can be solved by correctly keeping track of the symmetries of the low-lying excitations, something that can be done, but that goes beyond the scope of our present analysis.

## 5  Conclusion

We have explicitly constructed the ground state of a QFT obtained from the continuous limit of the Ising model as an MPS. We have done it by exploiting both Hastings's observation that the Euclidean time evolution generates a continuous MPS in imaginary time and the Lorentz invariance emerging at the critical point.

These two ingredients allow to characterize the QFT by studying the properties of the tMPS, a continuous MPS emerging, as a result of the Trotter expansion, along the temporal direction. As a result we have identified the QFT with the one first obtained by Zamolodchikov as the continuous limit of the Ising model perturbed with a weak magnetic field.

This result supports earlier claims [22,57] that the finite entanglement phenomenon is equivalent to perturb-

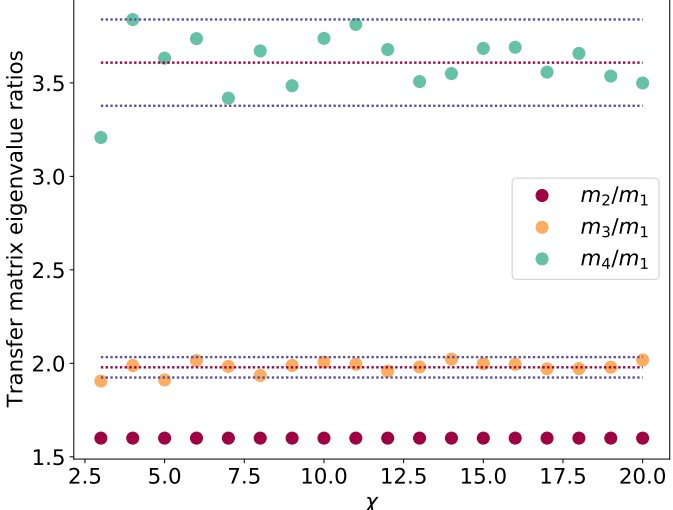

Figure 9: (Color online) The ratios of the eigenvalues of the temporal transfer matrix $E_T$ versus $\chi$ at $\tau = 10^{-6}$. We use it in order to identify the field theory that the tMPS describes. The lightest mass is fixed so to produce a ratio $m_2/m_1 = 1.6$. All the other ratios are computed accordingly. The present results seem to reproduce those in Ref. [60], thus allowing to identify the field theory encoded in the tMPS with the one obtained by weakly perturbing an Ising model with a magnetic field.

ing the critical Hamiltonian with the lightest relevant field with the correct symmetry properties. In particular in our case, since the $Z_2$ symmetry of the Ising model is broken in the finite entanglement regime, we can identify the relevant perturbation corresponding to the finite entanglement phenomenon with a parallel magnetic field.

Our identification, however, opens a new puzzle, since in our transfer matrix some of the expected excitations are absent. We leave the solution of this puzzle to further studies.

# Acknowledgements

We thank Ian McCulloch for stimulating discussions. L.T. acknowledges the discussion with Giuseppe Mussardo and Pasquale Calabrese on the mass spectrum of the perturbed Ising model and Andreas Läuchli and Philippe Corboz for discussions on related subjects. S.J.R. acknowledges Fundació Catalunya - La Pedrera · Ignacio Cirac Program Chair. We acknowledge the Spanish Ministry MINECO (National Plan 15 Grant: FISI-CATEAMO No. FIS2016-79508-P, SEVERO OCHOA No. SEV-2015-0522, FPI), European Social Fund, Fundació Cellex, Generalitat de Catalunya (AGAUR Grant No. 2017 SGR 1341 and CERCA/Program), ERC AdG OSYRIS, EU FETPRO QUIC, and the National Science Centre, Poland-Symfonia Grant No. 2016/20/W/ST4/00314.

# A  Algorithms

In this section, we briefly review the formulation of iDMRG [8, 16, 17] and iTEBD [6, 7]. In particular, we explain how to use these two algorithms to simulate the ground state of a 1D quantum system in the thermodynamic limit.

Moreover, we review the standard AOP [64, 65], and employing the idea of DMRG, we generalize the AOP for non-Hermitian problems. In this modified scheme, we show that two iMPSs appear along the two directions in the encoding of the 2D TN: one is the iMPS of iDMRG that represents an RG flow, and the other is the translationally invariant iMPS of iTEBD. In other words, the RG transformations in iDMRG are actually the

(a)

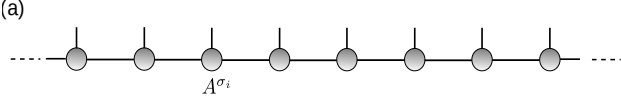

(b)

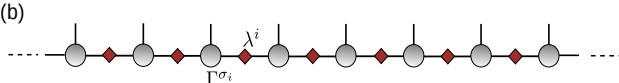

Figure 10: Graphical representation of an infinite matrix product state (MPS). In (a) the tensors $A^{\sigma_i}$ are in left canonical form (eq. (23)) (or right canonical form). In (b) we show the infinite matrix product state (MPS) in Vidal's formulation. $\Gamma$ is the local tensor while $\lambda$ represents the entanglement of the system.

truncations of the iMPS in iTEBD. It means that when iDMRG is implemented in one direction, the iTEBD is implemented at the same time in the other direction.

## A.1   Infinite Matrix Product State

A general quantum state $|\psi\rangle$ on a chain with $M$ sites can be written as

$$|\psi\rangle = \sum_{\sigma_1 \ldots \sigma_M} C_{\sigma_1 \ldots \sigma_M} |\sigma_1 \ldots \sigma_M\rangle, \tag{17}$$

with $C_{\sigma_1 \ldots \sigma_M}$ coefficients of the state and $|\sigma_j\rangle$ local basis. $C_{\sigma_1 \ldots \sigma_M}$ represents a high-rank tensor whose size increases exponentially with the number of sites.

To resolve such an "exponential wall", it has been proposed to write $C_{\sigma_1 \ldots \sigma_M}$ in an MPS form [3, 14, 18] that reads

$$|\psi\rangle = \sum_{\{\sigma\}} \sum_{\{\beta\}} A^{\sigma_1}_{1,\beta_1} A^{\sigma_2}_{\beta_1,\beta_2} \ldots A^{\sigma_M}_{\beta_{L-1},1} |\sigma_1 \ldots \sigma_M\rangle, \tag{18}$$

where $A^{\sigma_i}_{\beta_{i-1},\beta_i}$ is a third-order tensor, i.e., a $(\chi_{i-1} \times \chi_i)$ matrix for each value of $\sigma_i$ ($\chi_i$ the bond dimension of the virtual index $\beta_i$). Such a representation can be readily generalized to infinite systems with translationally invariant MPS, i.e. $A^{\sigma_l}_{\beta\beta'} = A^{\sigma}_{\beta\beta'}$ for $\forall \sigma_l$. (see Fig. 10).

Given a quantum state, its MPS representation is in general not unique, but it has gauge degrees of freedom. For each virtual bond $\beta_i$, we can define an invertible square matrix $X_i$, and rewrite the state Eq. (18) by inserting an identity in each virtual bond as

$$\ldots A^{\sigma_1}_{1,\beta_1} A^{\sigma_2}_{\beta_1,\beta_2} \ldots A^{\sigma_l}_{\beta_{l-1},\beta_l} \ldots = \ldots X_1 \left( X_1^{-1} A^{\sigma_2}_{\beta_1,\beta_2} X_2 \right) \ldots \left( X_{l-1}^{-1} A^{\sigma_l}_{\beta_{l-1},\beta_l} X_l \right) \ldots. \tag{19}$$

The new MPS is formed by $B$ tensor that is again a $d \times \chi_{l-1} \times \chi_l$ tensor satisfying

$$B^{\sigma_l} = X_{l-1}^{-1} A^{\sigma_l} X_l. \tag{20}$$

The MPS Eq. (18) is then written as

$$|\psi\rangle = \sum_{\{\sigma\}} \ldots B^{\sigma_1} B^{\sigma_2} \ldots B^{\sigma_l} \ldots |\sigma_1 \ldots \sigma_l \ldots\rangle, \tag{21}$$

which gives exactly the same state as that formed by $A$.

Therefore, the gauge of MPS is equivalent to the direct sum of the groups of isomorphisms of $\chi_l$ dimensioned complex vector spaces

$$\mathcal{G}_{MPS} = \oplus_{l=1}^{\infty} \text{Iso}\left(\mathcal{C}^{\chi}\right). \tag{22}$$

To fix the gauge, one can define the left-normalized form of the MPS, where the local tensor satisfies

$$\sum_{\sigma} A^{\sigma\dagger} A^{\sigma} = I. \tag{23}$$

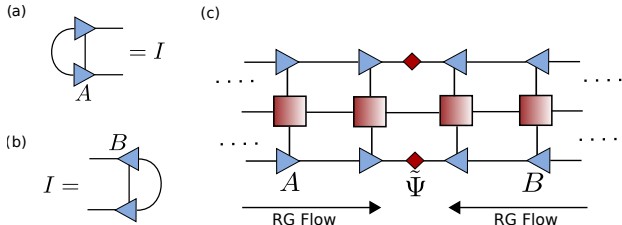

Figure 11: (Color online) The orthogonal conditions of the isometries (a) $A$ and (b) $B$. (c) The MPS that appears in DMRG is formed by $A$, $B$ and the central tensor $\tilde{\Psi}$. Such an MPS is in the orthogonal form and gives a RG flow in the physical space of the Hamiltonian $H$.

Similarly, the right-normalized form is defined as

$$\sum_{\sigma} B^{\sigma} B^{\sigma\dagger} = I. \tag{24}$$

Without loss of generality we can write $A^{\sigma_i}$ (or $B^{\sigma_i}$ ) as a product of $\Gamma$ complex tensor and a positive real diagonal matrix $\Lambda$. The MPS becomes (see Fig. 10 (b))

$$|\psi\rangle = \sum_{\sigma_1...\sigma_L} \Gamma^{\sigma_1}\Lambda^{[1]}\Gamma^{\sigma_2}\Lambda^{[2]}\cdots\Gamma^{\sigma_{l-1}}\Lambda^{[l-1]}\Gamma^{\sigma_l}\Lambda^{[l]}\cdots\Gamma^{\sigma_{L-1}}\Lambda^{[L-1]}\Gamma^{\sigma_L}|\sigma_1\cdots\sigma_L\rangle. \tag{25}$$

In this way, the MPS is written in canonical form and it is connected with the left and right orthogonal form by

$$A^{\sigma_l} = \Lambda^{[l-1]}\Gamma^{\sigma_l}, \quad B^{\sigma_l} = \Gamma^{\sigma_l}\Lambda^{[l]}. \tag{26}$$

In this form, one can prove that $\Lambda$ is actually the bipartite entanglement spectrum of the state.

## A.2 Infinite density matrix renormalization group

In the DMRG scheme, one typically starts with a short block (of length $l$) dubbed as the system (S) and its copy as the environment (E). The basis are denoted as $\{|\alpha_l^S\rangle\}$ and $\{|\alpha_l^E\rangle\}$, respectively. Then one grows the chain by adding two sites between the system and the environment. Then the total state can be written as

$$|\psi\rangle = \sum_{\alpha_l^S,\sigma_{l+1}^S,\sigma_{l+1}^E,\alpha_l^E} \Psi_{\alpha_l^S\alpha_l^E}^{\sigma_S\sigma_E}|\alpha_l^S\rangle|\sigma_{l+1}^S\rangle|\sigma_{l+1}^E\rangle|\alpha_l^E\rangle, \tag{27}$$

where $|\sigma_{l+1}^S\rangle$ and $|\sigma_{l+1}^E\rangle$ represent the basis of the sites added to the system and environment. The aim of the density matrix projection is to determine a subset of $\chi$ states $|\alpha_{l+1}^S\rangle$ ($|\alpha_{l+1}^E\rangle$) that optimally approximate the ground state of the enlarged system (environment) block. Accordingly, $|\alpha_{l+1}^S\rangle$ and $|\alpha_{l+1}^E\rangle$ are defined by the truncation matrices as

$$|\alpha_{l+1}^S\rangle = \sum_{\alpha_l^S} A_{\alpha_l^S\sigma_{l+1}^S,\alpha_{l+1}^S}|\alpha_l^S\rangle|\sigma_{l+1}^S\rangle, \tag{28}$$

$$|\alpha_{l+1}^E\rangle = \sum_{\alpha_l^E} B_{\alpha_l^E\sigma_{l+1}^E,\alpha_{l+1}^E}|\alpha_l^E\rangle|\sigma_{l+1}^E\rangle, \tag{29}$$

$A$ and $B$ are the isometries that realize the truncations of the basis, satisfying Eqs. (23) and (24) (see Fig. 11 (a) and (b)).

Then the ground state with truncated basis is written as

$$|\tilde{\psi}\rangle = \sum_{\alpha_{l+1}^S\alpha_{l+1}^E} \tilde{\Psi}_{\alpha_{l+1}^S\alpha_{l+1}^E}|\alpha_{l+1}^S\rangle|\alpha_{l+1}^E\rangle. \tag{30}$$

To minimize the truncation error, that is indicated by the quadratic cost function

$$S(|\tilde{\psi}\rangle) = \| |\psi\rangle - |\tilde{\psi}\rangle \|^2, \tag{31}$$

we use the truncation matrices given by the dominant eigenvectors of the reduced density matrix

$$\rho_S = \text{Tr}_\text{E}|\psi\rangle\langle\psi| = \Psi\Psi^\dagger, \tag{32}$$

$$\rho_E = \text{Tr}_\text{S}|\psi\rangle\langle\psi| = \Psi^\dagger\Psi. \tag{33}$$

Then, with eigenvalue decomposition one can obtain $A$ and $B$ as

$$\rho_S = AD^2A^\dagger, \quad \rho_E = BD^2B^\dagger. \tag{34}$$

One can easily see that $A$, $D$ and $B$ in fact give the optimal singular value decomposition (SVD) of $\Psi$ in Eq. (27), i.e.

$$\Psi_{\alpha_l^S \alpha_l^E}^{\sigma_A \sigma_B} \simeq \sum_{\alpha_{l+1}^S=1}^{\chi} A_{\alpha_l^S \sigma_{l+1}^S, \alpha_{l+1}^S} D_{\alpha_{l+1}^S} B_{\alpha_l^E \sigma_{l+1}^E, \alpha_{l+1}^E}, \tag{35}$$

showing that $D$ gives the entanglement spectrum of the ground state.

It is well-known that the ground state obtained by DMRG is actually an MPS formed by $A$, $B$ and $\tilde{\Psi}$. From Fig. 11 (c), one can see that the MPS is in an orthogonal form as introduced in the previous subsection, and gives a renormalization group (RG) flow of the physical Hilbert space. The direction of the RG flow is determined by the orthogonal conditions shown in Eqs. (23) and (24).

## A.3 Infinite time-evolving block decimation

In this section we will discuss about iTEBD, i.e. we will show how to update the iMPS for the state $|\psi\rangle$ by applying the operator $\hat{U} = e^{-i\hat{H}t}$:

$$|\psi(t)\rangle = e^{-i\hat{H}t}|\psi(0)\rangle, \tag{36}$$

where $t$ can be real or imaginary. Let us assume that $|\psi\rangle$ is written in MPS form as (25), and $\hat{H}$ is a nearest-neighbor Hamiltonian, i.e. $\hat{H} = \sum_i \hat{h}_i$ , where $\hat{h}_i$ contains the interaction between sites $i$ and $i + 1$. We can then discretize time as $t = N\tau$ with $\tau \to 0$ and $N \to \infty$ and use the Trotter-Suzuki decomposition, which approximates the operator $e^{-\hat{H}\tau}$. For example, the first order expansion reads:

$$\hat{U}(\tau) = e^{-i\hat{H}\tau} = e^{-i\hat{h}_1\tau}e^{-i\hat{h}_2\tau}\cdots e^{-i\hat{h}_L\tau} + O(\tau^2), \tag{37}$$

which contains an error due to the non-commutativity of bond Hamiltonians, $[\hat{h}_i, \hat{h}_{i+1}] \neq 0$. The second order expansion similarly reads:

$$e^{-i\hat{H}\tau} = e^{-\frac{i}{2}\hat{H}_\text{odd}\tau}e^{-i\hat{H}_\text{even}\tau}e^{-\frac{i}{2}\hat{H}_\text{odd}\tau}, \tag{38}$$

where we have to rewrite the Hamiltonian in the following way:

$$\hat{H} = \hat{H}_\text{odd} + \hat{H}_\text{even} = \sum_{i \in Z_{odd}} \hat{h}_i + \sum_{i \in Z_{even}} \hat{h}_i. \tag{39}$$

Therefore the evolution operator is expanded as a sequence of small two-size gates:

$$\hat{U}^{[i,i+1]} = e^{-i\hat{h}_{i,i+1}\delta\tau}, \tag{40}$$

which we arrange into two gates

$$\hat{U}_\text{even} = e^{-i\hat{H}_\text{odd}\tau} = \otimes_{i \in Z}\hat{U}^{[2i,2i+1]} \qquad \hat{U}_\text{odd} = e^{-i\hat{H}_\text{even}\tau} = \otimes_{i \in Z}\hat{U}^{[2i-1,2i]} \tag{41}$$

The updated step that is used in iTEBD is depicted in Fig 12 (a) and makes use of an SVD to obtain the new evolved tensors. To find the updated MPS, we repeat the application of gates $\hat{U}_\text{even}$ and $\hat{U}_\text{odd}$ on $|\psi\rangle$. In practice,

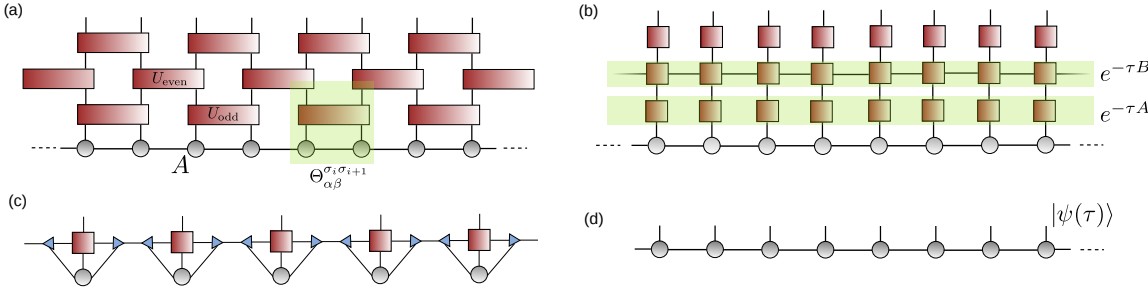

Figure 12: Graphical representation of $e^{-i\hat{H}\tau}|\psi(0)\rangle$. (a) The operator $e^{-i\hat{H}\tau}$ is expressed in the second order Trotter decomposition as a product of local gates. Then the gate $\hat{U}_{\text{odd}}$ (or $\hat{U}_{\text{even}}$) is contracted with two tensors of MPS in a single tensor $\Theta$. After the SVD decomposition of $\Theta$ we obtain the new tensors $A$ (b) The evolution operator is expressed in MPO language [75]. (c) Contraction of MPO and MPS. After the contraction the bond dimension of MPS grows to $\chi'$ and is brought back to $\chi$ by introducing a projector (blue triangle). (d) Final state $|\psi(\tau)\rangle$

we contract the gate $\hat{U}^{[2i,2i+1]}$ with two tensors of MPS in a single tensor $\Theta = A^{\sigma_i}\hat{U}^{\sigma_i,\sigma_{i+1}}A^{\sigma_{i+1}}$. Then we compute the SVD decomposition of $\Theta$ and recover the updated tensor $A^{\sigma_i}$ and $A^{\sigma_{i+1}}$ (see Fig. 12 (a)).

After the application of the Trotter gates the bond dimension of the bond under consideration grows to $\chi'$ and is back to $\chi$ by keeping only the largest $\chi$ singular values spectrum. Since the evolution is not unitary, canonicalization should be applied to reach the optimal truncations of the MPS [10]

Moreover it is also possible to write the imaginary time operator $\hat{U} = e^{-i\hat{H}\tau}$ by an infinite matrix product operator (iMPO) [19, 20]. This iMPO is formed by infinite copies of the tensor $\hat{T}^{\sigma_i,\sigma_i'}_{\mu,\nu}$, where $\sigma_i$ and $\sigma_i'$ are physical indices and $\mu$ and $\nu$ are bond indices

$$e^{-i\hat{H}\tau} = \hat{T}^{[1]}\hat{T}^{[2]}\cdots\hat{T}^{[l]}\cdots . \tag{42}$$

Following Ref. [75] it is possible to find the exact MPO of $U(\tau) = e^{-i\hat{H}\tau}$. Suppose that we want to find the ground state of the Ising Hamiltonian. The Hamiltonian is split into two parts $A$ and $B$ such that all terms within $A$ and within $B$ are commuting. For the Ising model $A = \sum_i \sigma_i^z\sigma_{i+1}^z$ and $B = \sum_i \sigma_i^x$. The Trotter decomposition reads

$$e^{A+B} = \lim_{n\to\infty}\left(e^{\frac{A}{n}}e^{\frac{B}{n}}\right)^n \tag{43}$$

The next step is to see how $e^{\tau A}$ and $e^{\tau B}$ look like. The latter is simple, as it is just equal to

$$e^{-i\tau B} = \otimes e^{-i\tau h\sigma_x}. \tag{44}$$

For the term $e^{-i\tau A}$, it is possible to demonstrate that there is a simple MPO description:

$$e^{-i\tau A} = \sum Tr(C_{\sigma_1}C_{\sigma_2}\cdots)X^{\sigma_1}X^{\sigma_2}\cdots \tag{45}$$

with $X^0 = I$ and $X^1 = e^{-\delta\tau\sigma_x}$.

Then the imaginary time operator $\hat{U} = e^{-i\tau\hat{H}} = e^{-i\tau B/2}e^{-i\tau A}e^{-i\tau B/2}$ is again exactly of the form 42 (see Fig. 12 (b)).

So $\hat{U}(\tau)$ for the Ising model is a simple MPO of bond dimension 2, and the evolution of $|\psi\rangle$ can be done in a very easy way: iteratively act the MPO on the MPS, and truncate to keep the bond dimension from being divergent (see the Fig. 12 (c-d)).

## A.4 Tensor network encoding

Several schemes have been developed to make approximate contractions of $2D$ TN [4, 13, 76]. In this section we review the AOP algorithm proposed in [64], in which a small few-body system is put in an entanglement bath to simulate the many-body system in the thermodynamic limit.

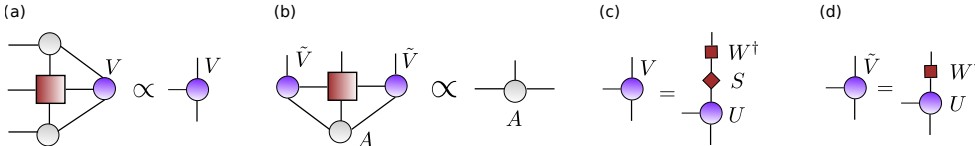

Figure 13: (Color online) The (a) and (b) show the two local eigenvalue equations given by Eqs. (48) and (49). The definition of $V$ and $\tilde{V}$ is shown by (c) and (d).

In the language of TN, AOP follows the idea that the contraction of an infinite TN is encoded into the contraction of the original local tensor with a proper boundary [77, 78]. The boundary is formed by tensors called boundary tensors. In other words, boundary tensors provide the entanglement bath that optimally mimics the entanglement between the local unit cell and the rest.

The boundary tensors, which can be randomly initialized, are determined by finding the fixed point of a set of local self-consistent eigenvalue equations. Then the infinite TN can be reconstructed (analogue to a "decoding" process) with the boundary tensors by utilizing the fixed-point conditions.

To show how AOP works, we consider a $2D$ tensor network with the structure shown in Fig. 1. Firstly, one chooses a super-cell, a finite block with $N$ spins and the cell tensor $T$ that is the local tensor of MPO obtained by Trotter-Suzuki decomposition of $\hat{U}(\tau) = e^{-\tau\hat{H}}$ (Eq. 42) [3]

Then we can define two transfer matrices along the spatial and time direction

$$E^S_{s_2b_1b_1',s_4b_2b_2'} = \sum_{s_1s_3} T_{s_1s_2s_3s_4} A^*_{s_1b_1b_2} A_{s_3b_1'b_2'}, \tag{46}$$

$$E^T_{s_1a_1a_2,s_3a_1'a_2'} = \sum_{s_2s_4} T_{s_1s_2s_3s_4} \tilde{V}^*_{s_2a_1a_1'} \tilde{V}_{s_4a_2a_2'}, \tag{47}$$

where $\tilde{V}$ is obtained by the SVD of dominant eigenstate of Eq. 46 $V$, i.e. $V = USW^\dagger$, $\tilde{V} = UW^\dagger$ (Fig. 13 (c)-(d)).

The tensor $\tilde{V}$ is introduced from $V$ in order to transform the non-local generalized eigenvalue problem to a local regular problem [Eqs. (46, 47)], where $E^T$ and $E^S$ are required to be Hermitian. The proof can be found in the Appendix in Ref. [64].

Then, the boundary tensors $A$ and $V$ are obtained as the dominant eigenstates of $E^T$ and $E^S$ (Fig. 13 (a)-(b)), respectively, i.e.,

$$\sum_{s'a_2a_2'} E^S_{sa_1a_1',s'a_2a_2'} V_{s'a_2a_2'} \propto V_{sa_1a_1'}, \tag{48}$$

$$\sum_{s'b_1'b_2'} E^T_{sb_1b_2,s'b_1'b_2'} A_{s'b_1'b_2'} \propto A_{sb_1b_2}. \tag{49}$$

One can see that such two eigenvalue problems are closely related to each other: one is parametrized by the solution of the other. [4]

When the boundary tensors simultaneously solve both equations, i.e. they converge to a self-consistent fixed point, the whole infinite TN is encoded into the local contraction of the cell tensor with the boundary tensors (Fig. 14).

Specifically speaking, we start with the local contraction (a scalar) given by

$$Z = \sum T_{s_1s_2s_3s_4} V^*_{s_2a_1a_1'} V_{s_4a_2a_2'} A^*_{s_1a_1a_2} A_{s_3a_1'a_2'}. \tag{50}$$

---

[3]The tensor $T$ can be found as suggested in [79].

[4]The eigenvalue equation (48) may not have a solution due to the block-triangular structure of the MPO of $\hat{H}$. This problem can be overcome quite easily for local Hamiltonians [15, 41, 81].

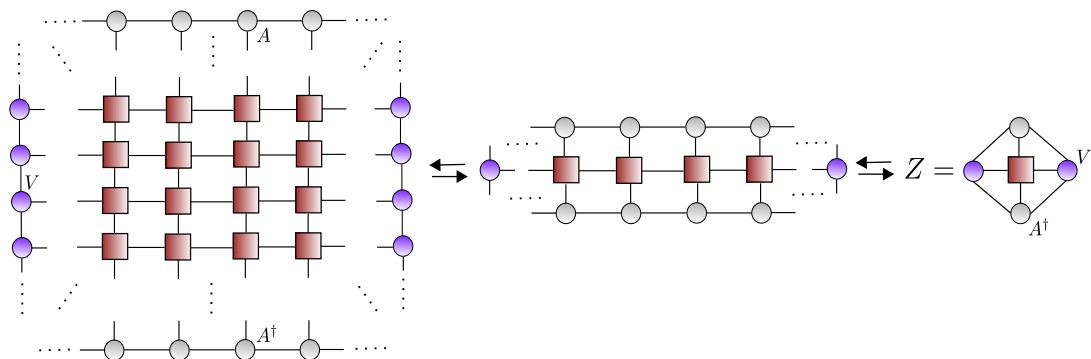

Figure 14: (Color online) The illustration of TN "encoding" (leftward) and "decoding" (rightward). One starts from an infinite TN formed by the cell tensor $T$ with two MPSs and their conjugates defined on the boundaries along the two directions of TN. Using the eigenvalue equations (48) and (49), respectively, the TN contraction is firstly transformed into a 1D TN and then finally into a local contraction given by Eq. (50). By going rightward, the infinite TN can be reconstructed from the local function, analog to a decoding process.

The eigenvalue equations indicate that $Z$ is maximized by the boundary tensors. Then, we repeatedly do the substitution with Eq. (48) and Eq. (49). It should be noted that the normalization of $V$ is required here as a constraint. Then Eq. (50) is equivalent to construct the partition function $Z$ defined as:

$$Z = \langle\psi|\hat{\rho}|\psi\rangle, \tag{51}$$

with $|\psi\rangle$ an infinite MPS formed by the tensor $A$ and $\hat{\rho}$ an infinite MPO formed by $T$.

Then, one utilizes the fact that $A$ is the solution of the maximization of $Z$. It means that the MPS $|\psi\rangle$ formed by $A$ maximizes Eq. (51), i.e., $|\psi\rangle$ is the ground state of the MPO. In other words, we have a (non-local) eigenvalue equation

$$\hat{\rho}|\psi\rangle \propto |\psi\rangle, \tag{52}$$

under the assumption that the ground-state of $\hat{\rho}$ can be effectively represented as an MPS. Eq. (52) is true because Eq. (51) is maximized by $|\psi\rangle$ while one has $\langle\psi|\psi\rangle = 1$ guaranteed by the equations shown in Figs. 13. By substituting Eq. (52) in Eq. (51) repeatedly, the infinite TN formed by $T$ can be reconstructed, meaning the whole TN is encoded into the local contraction given by Eq. (50).

The above analyses suggest that an MPS defined by $V$ emerges in the vertical direction (Fig. 14). This in fact defines the tMPS. In the next section, we will show more explicitly by using iDMRG and iTEBD that this MPS gives the dominant eigenstate of the vertical transfer matrix of the TN. In other words, the tMPS is the boundary MPS of the TN.

In practice, the AOP is implemented in the following steps:

- **Step 1:** From the Hamiltonian $\hat{H}$ define the local tensor $T$.

- **Step 2:** Give an initial guess of the tensor $A$. While its elements can be totally random, it is better to make the two ancillary indexes symmetrical so that $E^S$ could be Hermitian from the beginning.

- **Step 3:** Calculate $E^S$ and its dominant eigenstate $V$ which is actually a third-order tensor. To compute this eigenvalue problem, one can use $V$ obtained from the last iteration as the initial guess.

- **Step 4:** Calculate $\tilde{V}$ from $V$, and the time transfer matrix $E^T$; solve its eigenstate $A$. Again, one can use the $A$ obtained in the last iteration as the initial guess.

- **Step 5:** Check if $A$ converges. If it does, proceed to Step 6; if not, go back to Step 3.

- **Step 6:** Use the tensor $A$ to construct the spatial MPS and $V$ for the tMPS; calculate the interested physical quantities.

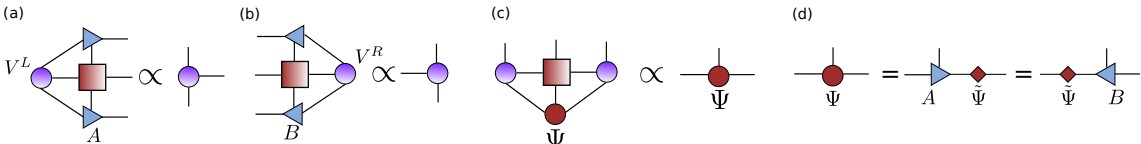

Figure 15: (Color online) The (a), (b) and (c) show the three local eigenvalue equations given by Eqs. (53, 54, 55). The isometries $A$ and $B$ are obtained by the QR decompositions of $\Psi$ in two different ways, as shown in (d).

## A.5 Generalized tensor network encoding

In this section we generalize the AOP scheme where the tMPS naturally emerges, and we show that the iDMRG and iTEBD can be unified.

Here, we take the one-site iDMRG as an example. As shown in Fig. 15 (a), (b) and (c), one can similarly write three local eigenvalue equations, which are given by three matrices

$$E^{S,L}_{s_2 b_1 b'_1, s_4 b_2 b'_2} = \sum_{s_1 s_3} T_{s_1 s_2 s_3 s_4} A^*_{s_1 b_1 b_2} A_{s_3 b'_1 b'_2}, \tag{53}$$

$$E^{S,R}_{s_2 b_1 b'_1, s_4 b_2 b'_2} = \sum_{s_1 s_3} T_{s_1 s_2 s_3 s_4} B^*_{s_1 b_1 b_2} B_{s_3 b'_1 b'_2}, \tag{54}$$

$$E^{T}_{s_1 a_1 a_2, s_3 a'_1 a'_2} = \sum_{s_2 s_4} T_{s_1 s_2 s_3 s_4} V^{L}_{s_2 a_1 a'_1} V^{R}_{s_4 a_2 a'_2}. \tag{55}$$

$V^L$, $V^R$ and $\Psi$ are the eigenstate of these three matrices that respectively satisfy

$$\sum_{s b_1 b'_1} V^{L}_{s b_1 b'_1} E^{S,L}_{s b_1 b'_1, s' b_2 b'_2} \propto V^{L}_{s' b_2 b'_2} \tag{56}$$

$$\sum_{s' b_2 b'_2} E^{S,R}_{s b_1 b'_1, s' b_2 b'_2} V^{R}_{s' b_2 b'_2} \propto V^{R}_{s b_1 b'_1} \tag{57}$$

$$\sum_{s' a'_1 a'_2} E^{T}_{s a_1 a_2, s' a'_1 a'_2} \Psi_{s' a'_1 a'_2} \propto \Psi_{s a_1 a_2}. \tag{58}$$

The matrices in Eqs. (53) and (54) do not need to be Hermitian.

$A$ and $B$, which are the left and right orthogonal part of $\Psi$, are obtained by QR decomposition [Fig. 15 (d)] as

$$\Psi_{s a_1 a_2} = \sum_{a'} A_{s a_1 a'} \tilde{\Psi}_{a' a_2} = \sum_{a'} \tilde{\Psi}^\dagger_{a_1 a'} B_{s a' a_2}, \tag{59}$$

where $A$ and $B$ are isometries, satisfying orthogonal conditions as

$$\sum_{sa} A_{s a a_1} A^\dagger_{s a a_2} = I_{a_1 a_2}, \tag{60}$$

$$\sum_{sa} B_{s a_1 a} B^\dagger_{s a_2 a} = I_{a_1 a_2}. \tag{61}$$

Now we interpret iDMRG by local self-consistent eigenvalue equitations. $V^L$ and $V^R$ represent the system and environment super-block. $E^T$ can be considered as the effective Hamiltonian (note that instead of using the MPO of Hamiltonian, here in AOP, we use the operator $e^{-\tau \hat{H}}$) with $\Psi$ its ground state. By the QR decompositions on $\Psi$, the renormalization of the basis of the system and environment are given by $A$ and $B$, and $\tilde{\Psi}$ is the center matrix in iDMRG [19,20].

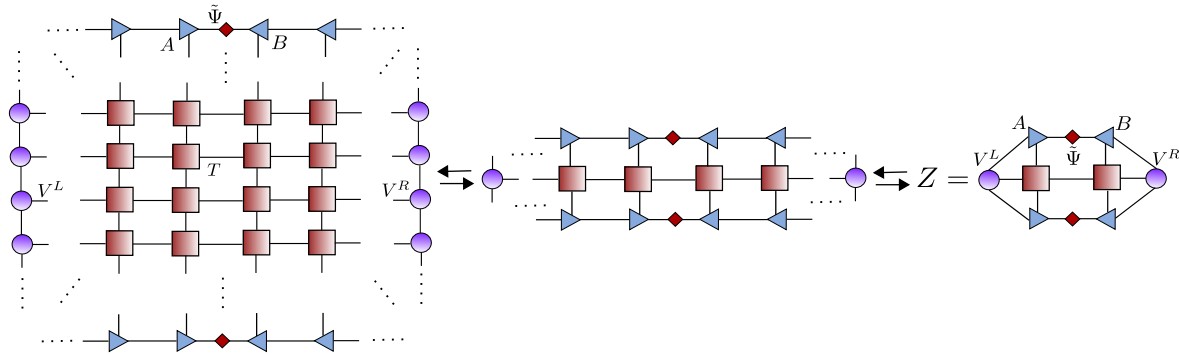

Figure 16: (Color online) The TN encoding given by iDMRG, where the TN contraction is equivalently transformed into a 1D TN and then finally into a local contraction.

Similarly to the AOP, the original TN can be reconstructed by repeatedly using the eigenvalue equations. We start from a local contraction (a scalar $Z$) of $T$ and the boundary tensors as shown on the right of Fig. 16. Then with Eqs. (56) and (57), $Z$ is transformed to the product of an MPO with an MPS and its conjugate. One can see that like iDMRG, the MPS is formed by $A$'s on the left side (left-orthogonal part) and $B$'s on the right side (right-orthogonal part) with the center matrix $\tilde{\Psi}$ in the middle. Then by using the fact that such an MPS optimally gives the ground state, we can use the corresponding eigenvalue equation to reconstruct the entire TN (Fig. 16).

In the encoding, there are three constraints: the normalizations of $V^L$, $V^R$ and the spatial MPS. The first two constraints are obviously local with $\langle V_L | V_L \rangle = 1$ ($\langle V_R | V_R \rangle = 1$). For the third one, by utilizing the orthogonal conditions of $A$ and $B$, the normalization of the MPS is equivalent to that of $\Psi$, which is also local. Then the equations like those in Fig. 15 (c) and (d) are no longer needed in this case. Thus, all eigenvalue problems are local and regular.

In the above scheme, one can explicitly see that $V^L$ (or $V^R$) also gives an MPS, and interestingly, such an MPS is updated in the way of iTEBD contracting along the spatial direction.

Let us pay attention to the eigenvalue equation that $V^L$ satisfies Eq. (56). If one chooses to solve it using a power method, i.e. to find the ground state of $E^{S,L}$ by updating $V^L$ with the product $V^L E^{S,L}$, such a product is equivalent to evolving the local tensor of the MPS in iTEBD with the MPO formed by $T$ in the vertical direction. In the evolution, the bond dimension of the local tensor $V^L$ increases exponentially. Then, truncation is implemented by contracting with the isometry $A$.

The TN encoding shows that while implementing iDMRG to get the MPS along the real-space direction, one is actually implementing iTEBD that gives the tMPS along the imaginary-time (or vertical) direction.

Some interesting discussions about the relations between (real-space) DMRG and the optimization of the real-space MPS were given [80]. These relations can be further applied to the tMPS using our proposal.

Note that $A$ is obtained from the QR decomposition of $\Psi$. Considering the eigenvalue equation Eq. (58), $\Psi$ in fact represents one half of the infinite environment of the vertical MPS formed by $V^L$, thus $A$ gives the optimal truncation matrix. (Obviously, the same discussion can directly apply to B and $V^R$)

Considering that the evolution of $V^L$ in the language of iTEBD is non-unitary, one important difference between the AOP and the iTEBD contracting in the spatial direction [63] is that the explicit implementation of canonicalization [10] is no longer necessary thanks to the eigenvalue equation Eq. (58).

The algorithm is implemented in the following steps:

- **Step 1:** From the Hamiltonian $\hat{H}$, define the local tensor $T$; give an initial guess of the tensor $\Psi$.

- **Step 2:** Calculate $E^{S,L}$ and $E^{S,R}$; solve their dominant left and right eigenstates $V^L$ and $V^R$, respectively. One may use the tensors obtained from the last iteration as the initial guess. Since $E^{S,L}$ and $E^{S,R}$ are not Hermitian, we update $V^L$ and $V^R$ as $V^L \leftarrow V^L (E^{S,L})^n$ and $V^R \leftarrow (E^{S,R})^n V^R$. The algorithm exactly becomes iDMRG (as we argued above) when taking the integer $n = 1$. It is also allowed to take

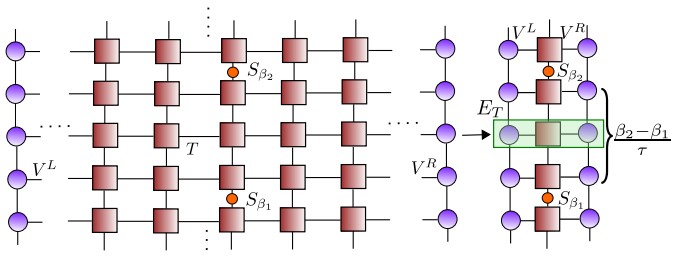

Figure 17: (Color online) The TN representation of the dynamic correlation function $\mathcal{C}(d\beta) = \langle \hat{S}_{\beta_1} \hat{S}_{\beta_2} \rangle$ with $d\beta = \beta_2 - \beta_1$ ($\beta_1 < \beta_2$), which can be calculated by the contraction of the tMPSs with an infinite TN stripe in the vertical direction.

$n > 1$, which speeds up the convergence. But possible instability may appear as $n$ increases due to the non hermitian effective Hamiltonian.

- **Step 3:** Calculate $E^T$ and its eigenstate $\Psi$. Again, one can use the $A$ obtained in the last iteration as the initial guess. Obtain $A$ and $B$ by the QR decompositions of $\Psi$.

- **Step 4:** Check if $A$ converges. If it does, proceed to Step 5. If not, go back to Step 2.

- **Step 5:** Use the tensors $\Psi$, $A$ and $B$ to construct the spatial MPS, and $V^{L[R]}$ for the tMPS; calculate the interested physical quantities, such as energy and entanglement.

# B  Calculations of correlation length

In the following, we explain how the correlation length of a translationally invariant MPS is calculated from the eigenvalues of the transfer matrix. This trick has been used frequently.

Let us explain with TN the calculation of $\mathcal{C}(d\beta) = \langle \hat{S}_{\beta_1} \hat{S}_{\beta_2} \rangle$ with $d\beta = \beta_2 - \beta_1$ ($\beta_1 < \beta_2$), where the two spin operators are put in the same site but at different imaginary time $\beta_1$ and $\beta_2$. Fig. 17 gives its TN representation (up to a normalization factor). From the argument with iTEBD, we know that the tMPS is the dominant eigenstate of the MPO defined by an infinite stripe of the TN in the vertical direction, thus, $\langle \hat{S}_{\beta_1} \hat{S}_{\beta_2} \rangle$ can be simplified as the contraction of the tMPSs with the MPO (right side of Fig. 17).

Now it is clear to see that $\langle \hat{S}_{\beta_1} \hat{S}_{\beta_2} \rangle$ is the matrix product of infinite $E_T$'s and two spin operators: there are $K = (\beta_2 - \beta_1)/\tau$ of $E_T$'s between these two operators with infinite $E_T$'s on both sides. It means that the decay of $\mathcal{C}(d\beta)$ is dominated by the gap between the first and second eigenvalues (denoted by $\eta_0$ and $\eta_1$) of $E_T$, i.e.,

$$\mathcal{C}(d\beta) \propto \left[ \frac{\eta_1}{\eta_0} \right]^K. \tag{62}$$

Here we take into account the fact that at the quantum critical point, $\langle \hat{S} \rangle$ still vanishes, thus $\mathcal{C}(\infty) = 0$. Then, one can readily obtain the dynamic correlation length

$$\tilde{\xi}_T = \frac{1}{\tilde{\Delta}}, \tag{63}$$

with $\tilde{\Delta} = \ln \eta_0 - \ln \eta_1$ the logarithmic gap of $E_T$ and $\tilde{\xi}_T = \xi_T \tau$ where $\xi_T$ is the correlation length calculated directly to the tMPS. The dynamic correlation function is known to possess a linear relation with the inverse of the system gap $\Delta$ [82], satisfying $\tilde{\xi}_T \propto 1/\Delta$. Then, we have the simple relation between the logarithmic gap of $E_S$ and the gap of the model as

$$\Delta \propto \tilde{\Delta}. \tag{64}$$

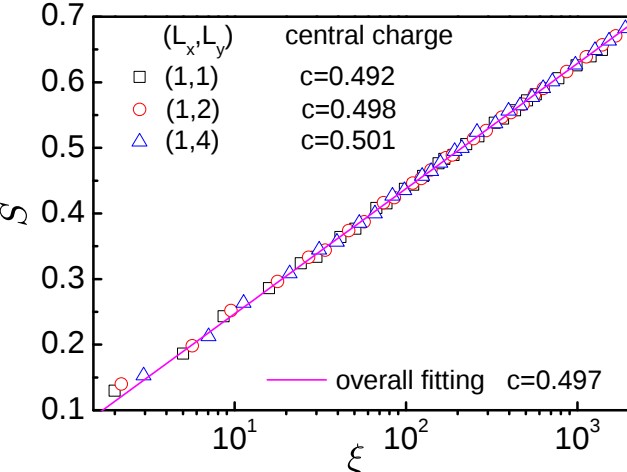

Figure 18: (Color online) The temporal entanglement entropy S as a function of the correlation length $\xi$ at the critical temperature. Each point is obtained by choosing the bond dimension $\chi = [2, 50]$ and the size of the unit-cell $(L_x, L_y)$. A fit to the expected logarithmic scaling allows to extract $c = 0.5$ as expected. The results from the spatial MPS are the same as those from the tMPS with the difference $O(10^{-5})$.

## C  Two-dimensional classical Ising partition function

In this section, we apply the TN encoding scheme to 2D classical Ising model at the criticality. There are some intrinsic differences between this TN of 2D classical partition and that of the (1+1)D quantum theory. For the 2D classical partition, the two dimensions of the TN are both spatial and discrete, and they are equivalent to each other.

Its partition function can be directly written in a 2D TN, where the local tensor is the probability distribution of some local Ising spins. Here, we take the Ising model on square lattice as an example, where the local tensor is defined as

$$T_{s_1 s_2 s_3 s_4} = e^{-\beta(s_1 s_2 + s_2 s_3 + s_3 s_4 + s_4 s_1)}, \tag{65}$$

with $\beta$ the inverse temperature and the spin index $s_i = \pm 1$. Note that the local tensor of the TN can also be chosen as the contraction of several $T$'s.

It is well-known that the dominant eigenstate of the transfer matrix can be approximated as an MPS. Each MPS with a finite bond dimension $\chi$ corresponds to a gapped state with a finite correlation length $\xi$ and entanglement entropy $S$. At the critical temperature, the central charge can be extracted by the scaling behavior of $S$ against $\xi$ [21–23]. Specifically speaking, with different $\chi$, $\xi$ and $S$ satisfy

$$S = \frac{c}{6} \ln \xi + const, \tag{66}$$

where the coefficient gives the central charge $c$. Note that Eq. (66) is independent of calculation parameters. Meanwhile, one can also check the scaling behavior with different $\chi$'s separately and have

$$\xi \propto \chi^{\kappa}, \tag{67}$$
$$S = \frac{c\kappa}{6} \ln \chi + const. \tag{68}$$

By substituting, one can readily have Eq. (66) from these two equations. For the 2D Ising model, we have the critical temperature $\beta_c = \ln(1 + \sqrt{2})/2$ from the exact solution [83] and $c = 1/2$ that corresponds to a free fermionic field theory [21]. We shall stress that for any finite $\chi$, we cannot exactly give a critical state by MPS,

but only a gapped. The central idea of the scaling theory with MPS is to extract the conformal data from the scaling behaviors the gapped MPSs.

In Fig. 18, we show that the MPS from iDMRG has the same correlation length $\xi$ and entanglement entropy $S$ as the vertical MPS obtained simultaneously in the iTEBD in the other direction with the difference $O(10^{-5})$. In other words, these two MPSs, though updated within two different schemes and located in two different directions of the TN, are connected by a gauge transformation. By choosing different bond dimension cut-offs and cells to construct the tensor $T$, the relation between $\xi$ and $S$ shows a robust logarithmic scaling, giving accurately the central charge. The precision increases with the size of the cell tensor $L_x$ and $L_y$.

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
