# Peer review of "Characterizing the quantum field theory vacuum using temporal Matrix Product states"

_SciPost Physics_

## Round 2 · Referee Report · Anonymous · 2018-11-29

Strengths

1- first explicit construction of the temporal continuous MPS corresponding to the limit of a lattice model.
2- the method could in principle be applied to other models.

Weaknesses

1- The writing can be considerably improved. Some aspects require extended explanation. Specially in the appendix there are numerous typos and grammar mistakes, to the point that sometimes the logical flow is obscured.
2- The captions of figures in the main text are not all clear.
3- Numerical results seem weak. The largest bond dimension is Xi=20, which is very small, considering in particular that the gaps shown in Fig. 9 are not converged. Also the effect of the unit cell L seems important, but there is no discussion about its influence on the conclusions.

Report

The paper studies the continuous limit of a MPS arising from a tensor network representation of the ground state of a lattice model, concretely the transverse field Ising Hamiltonian. The ground state is expressed as an infinitely long imaginary time evolution, which is discretized in Trotter steps. The work builds on an observation by Mahajan and Hastings in Phys. Rev. A 91, 032306 (2015), where it was noticed that the fixed point of the transfer operator of this network gives a continuous MPS in the limit of vanishing Trotter step.

The authors find explicitly this cMPS and identify it as the ground state of a 1+1D quantum field theory. In the critical case, in spite of the space-time symmetry that should connect the MPS in spatial and temporal directions, they find that the temporal MPS corresponds to a different theory, and argue that the reason is the finite bond dimension of the state.

The paper is interesting and opens possibilities for further investigation, but the presentation of the work should be improved before publishing.

In particular, the main text should contain a clear explanation of which algorithms are used. Are all the results obtained using AOP? Acronyms (in special AOP) need to be defined, and the essence of this method, or at least the role of the unit cell should also be presented in the text. Also, why is a unit cell of length L=4 (e.g. in Fig. 3) enough?

Regarding the explicit continuum limit of the MPS tensors, does the choice of gauge in the discrete tensors not affect the identification of the components from which the continuum limit is obtained?

The results in Fig. 9 do not seem completely conclusive. Considering that the first mass ratio is fixed, the second one is found to be compatible with Zamolodchikov’s result, but oscillating around this value, and the third one shows larger deviations. It is not clear that the ratios are converging when the bond dimension increases. Results with larger Xi would be necessary. Also, could it happen that different mass ratios appear in the spectrum for larger Xi?

Requested changes

Besides what is discussed in the Report above:

1- On page 5, the caption of figure 3 does not explain panels c and d.

2- At the beginning of section 3, "the parallel MPS" should probably be "horizontal".

3- The statement that the properties of tMPS where only studied in Ref. [48] is not correct: entanglement properties where studied in [New J. Phys. 14 (2012) 075003] and later for a modified construction in [Phys. Rev. B 89, 201102 (2014)].

4- The caption of Fig. 4 should contain a more clear explanation of what is plotted (e.g. is it all the components? Are there not 20x20 per tensor? Are they all real?)

5- On page 7, Eq. (7) has some typos (sigma for n, L for M).

6- On page 7, in Eq. (8), the proper factor should be sqrt(tau) (and correspondingly in Eq. (9) ) for dimensional consistency (see e.g. ref. [35]). In Fig. (4) that actually appears to be the slope.

7- On page 8, what is the logical relation between Eq. (12) and (13)? Does (12) not follow from (13)?

8- On page 9, around Eq. (15), an explicit explanation of the super- or subindices n is missing (presumably referring to the different eigenstates)

9- The text of the appendix should be revised for grammar and typos.

---

## Round 2 · Referee Report · Anonymous · 2019-1-3

Strengths

This paper poses two interesting research questions
1-What is the physics of the fixed point of the quantum transfer matrix obtained from a Trotter decomposition of the imaginary time evolution. This is an interesting question both for gapped and critical systems.
2-What is the physical meaning of regulation by finite bond dimension / finite entanglement in critical systems?

Weaknesses

Unfortunately, these two questions are not properly addressed because of
1-limited numerics (small bond dimension, only Ising model)
2-various confusing, limited or ill-posed descriptions (see examples below)

Report

In this paper, the authors aim to study the continuous MPS fixed point along the time direction of a Trotterized imaginary time evolution for a one-dimensional quantum spin chain, called the tMPS. Gaining better insight into the structure of this tMPS is an interesting topic, both for gapped and critical systems. The authors choose to only study the question for critical systems, and at the same time use this study as a means of gaining insight into the role of finite entanglement/bond dimension as a QFT regulator. These two questions seems to get mixed up, and a clear answer to neither of them is in the end obtained. For example, does one now understand whether the tMPS is the (approximate) ground state of some QFT Hamiltonian that can explicitly be written down?

At various places throughout the text, it seems to me that the authors mistake Lorentz invariance in the low energy region of a critical point with full blown Lorentz invariance of a model. Whereas the low energy behavior of the physical MPS (defined on a lattice system) and the tMPS (defined in the continuum) is related by Lorentz invariance, the UV behavior of both systems can be very different. I don't understand the quantitative meaning or implication of statements such as "the tMPS and the standard ground sate MPS, at the critical point, are related by a trivial non-universal re-scaling factor".

A similarly confusing statement is: "... the system becomes not only scale invariant but Lorentz invariant (this is equivalent to Galileo invariance in Euclidean time)." By definition, a Galileo transformation is very different from a Lorentz transformation, so I do not understand the meaning of this sentence.

A slightly more accurate description is in terms of the anisotropic classical Ising model. It is true that the imaginary time evolution of the quantum transverse Ising model is related to a highly anisotropic limit of the classical ising model in 2d. But then, in discussing the scaling factor $\nu$, the authors are comparing the physical temporal correlation $\tilde{\xi}_T = \xi_T \tau$ with the spatial correlation length of the lattice model $\xi$ in terms of lattice units. So whereas one has the units of time, the other is dimensionless, and the "scaling factor" $\nu$ is not a dimensionless number. It is thus not a "non-universal rescaling of the velocity of excitations" but truly a velocity (in terms of lattice distance) in itself. This whole section is formulated very sloppily and should be revised.

Other very confusing paragraphs damage the validity of the paper severely:

* Below eq 4, the Ising model $H= J \sum_i S^x_i S^x_{i+1} - h \sum_i S^z_i$ is described to acquire a non-zero value for the order parameter $<S^z>$ in the paramagnetic phase, and an expectation value that goes to zero the ferromagnetic phase? With the above Hamiltonian for the transverse field ising model, the order parameter is $S^x$, not $S^z$, and its expectation value will be non-zero in the ferromagnetic phase (going to zero at the critical point) and strictly zero in the paramagnetic phase.

* The rescaling of the masses at the end of section 4: it is stated that $m_1$ is fixed by normalization and then the ration $m_2/m_1$ is further adjusted to a specific value? Normally, the eigenvalue $\tilde{\eta}_0$ is fixed by normalization, and this also fixes all the other eigenvalues of the transfer matrix. For the tMPS, it should be that the eigenvalues of the transfer matrix are of the form $\eta_n = \eta_0 \exp(- \tau \tilde{m}_n)$ and so all the $\tilde{m}_n$ are completely fixed for a given bond dimension. They might vary as a function of bond dimension, but then the ratios $m_2/m_1$ should at least converge to a fixed value, if any of the following discussion is supposed to make sense. So I don't see how anything can be "adjusted" in this analysis?

* The dependence on the unit cell of the AOP scheme? The reader is referred to another paper for this, but a short summary of the findings should be presented. In particular, I don't understand how the unit cell of the numerical scheme can have such a large impect on the extracted quantities, given that the physical state for the lattice system is supposed to be translation invariant.

Finally, some statements are completely beside the point, e.g. the fact that the $Q$ and $R$ matrices of the tMPS are simply obtained from the scaling behavior and thus "overcome the difficulties that arise when trying to directly optimize the continuous MPS". Indeed, this is a completely different setting so how is this even related. It would be, if one would be able to tell if this tMPS is the approximate ground state of a certain QFT Hamiltonian, which goes back to my first question.

Other questions that I might have: could one impose Z2 symmetry in the simulation? How would this affect the results, i.e. would one still find the universal mass ratios of the Zamalodchikov QFT?

Requested changes

I believe this paper needs a major revision in order to be appropriate for publication. I've pointed out several confusing or plainly wrong paragraphs in the report. These should definitely be corrected, but a more significant revision in which the authors clearly lay out their research questions and corresponding findings, supported by additional numerics (e.g. a different model) might be required.

---

## Editorial Decision

editor-in-charge_assigned